# Hydrophobic Organic Pollutants in Soils and Dusts at Electronic Waste Recycling Sites: Occurrence and Possible Impacts of Polybrominated Diphenyl Ethers

**DOI:** 10.3390/ijerph16030360

**Published:** 2019-01-28

**Authors:** Chimere May Ohajinwa, Peter M. Van Bodegom, Qing Xie, Jingwen Chen, Martina G. Vijver, Oladele O. Osibanjo, Willie J.G.M. Peijnenburg

**Affiliations:** 1Institute of Environmental Sciences (CML), Leiden University, P.O. Box 9518, 2300 RA, Leiden, The Netherlands; p.m.van.bodegom@cml.leidenuniv.nl (P.M.V.B.); vijver@cml.leidenuniv.nl (M.G.V.); 2Key Laboratory of Industrial Ecology and Environmental Engineering (Ministry of Education), School of Environmental Science and Technology, Dalian University of Technology, Dalian 116024, China; qingxie@dlut.edu.cn (Q.X); jwchen@dlut.edu.cn (J.C); 3Department of Chemistry, University of Ibadan, Ibadan 200282, Nigeria; oosibanjo@yahoo.com; 4Center for Safety of Substances and Products, National Institute of Public Health and the Environment (RIVM), P.O. Box 1, 3720 BA, Bilthoven, The Netherlands

**Keywords:** electronic waste, informal recycling, PBDEs, soil, dust, Nigeria

## Abstract

Concerns about the adverse consequences of informal electronic waste (e-waste) recycling is increasing, because e-waste contains some hazardous substances such as polybrominated diphenyl ethers (PBDEs) which is used as flame retardants in electronics. There is dearth of information on the concentrations of PBDEs and the pattern of distribution at the various e-waste recycling sites in Nigeria. This study therefore measured the concentrations of 13 PBDE congeners, in top soils (0–10 cm) and in various dust samples from different e-waste recycling sites (burning, dismantling, repair). PBDE concentrations at e-waste sites were compared with the concentrations in samples from corresponding control sites in three study locations in Nigeria (Lagos, Ibadan, and Aba). There were significant differences in the level of PBDEs congeners between each of the e-waste recycling sites and the corresponding control sites. The levels of PBDEs at the e-waste recycling sites exceeded the levels at the controls sites by a factor of 100 s to 1000 s. In general, PBDE concentrations at the e-waste sites decreased with the intensity of the e-waste recycling activities: burning sites > dismantling sites > repair sites > control sites. Our results suggest that the informal e-waste recycling has negative impacts on the enviroment and human health.

## 1. Introduction

Across the globe, electronic or electrical devices have become indispensable in our daily lives and the use of electronic electrical device is growing at great speed. It is characterized by an increasing number of users and rapid technological advances driven by efficiency, social and economic development. Many people now own multiple personal electronic devices such as information and communication technology (ICT) devices, but the life span of these devices are getting shorter mainly because they become obsolete more quickly compared to the past. In addition, most of these devices are disposed even before they become dysfunctional so as to make space for newer devices with better specifications/functions. Therefore, exponentially growing demand for electronic equipment has led to a rapid increase in the rate of electronic waste (e-waste) generated. e-Waste, also known as Waste Electrical and Electronic Equipment (WEEE), is one of the fastest growing municipal waste streams [1]. In 2016, 44.7 million metric tonnes (Mt) of e-waste were generated globally, and this amount is expected to increase to 52.2 million metric tonnes by 2021 [2].

The concern about e-waste is not only about the volumes generated but also about the unsafe methods used in recycling the electronics, mainly in developing countries. It is reported that 80% of e-waste generated globally is recycled informally or simply dumped at dumpsites or landfils in developing counties in Asia and in Africa [1,2]. The informal/unsafe methods of managing e-waste in developing countries perpetuate due to absence of infrastructure for appropriate waste management, an absence of end-of-life product take-back system, or implementation of extended producer responsibility (EPR) schemes by manufacturers which is enforced in developed countries [1,2]. In addition EPR has been extended beyond the e-waste sector to include packaging waste generally in developed countries [3]. e-waste contains several different substances, some of which are compounds of potential concern which include products of incomplete combustion, reformation products after combustions, elements such as metals-lead, mercury, cadmium, arsenic, beryllium, and flame retardants such as polybrominated diphenyl ethers (PBDEs) [4,5]. These mixtures of different substances, covering both chemicals present in EEE components and mixtures of chemicals released during e-waste processing, may pose significant implications for human health and environmental safety [6,7].

PBDEs are a class of persistent organic pollutants (POPs) which have been used as flame retardants in many consumer products such as electronic equipment, textiles, furniture, automobile seats and other consumer products since the 1970s. PBDEs are also a class of additive brominated flame retardants (BFRs), which are not covalently bound to the products (polymer matrices). In case of a fire, bromine radicals are released as a result of thermal energy. These radicals decrease the flame, and they reduce heat and carbon monoxide production. Because the PBDEs are not permanently bound to the polymer matrices, they are widely dispersed in the environment. In total, PBDE has 209 congeners, which are dependent on the number and position of the bromine atoms on the two-phenyl rings. Approximately 56,418 metric tons of PBDEs were produced globally in 2003 [8]. PBDEs have mostly been produced and used in three commercial groups: pentabromodiphenyl ether (penta-BDE; C_12_H_5_Br_5_O), octabromodiphenyl ether (octa-BDE; C_12_H_2_Br_8_O), and decabromo- diphenyl ether (deca-BDE; C_12_Br_10_O) with about 11%, 6%, and 83% of global PBDEs respectively [8,9]. Penta- and octa-BDEs are not single chemicals. Penta-BDE comprises tri-, tetra-, penta-, and hexa-BDE; Octa-BDE comprises hexa-, hepta-, and nona-BDEs; and deca-BDE is a single chemical (BDE 209). The major sources of PBDEs are the manufacturing sector, product application, recycling processes, thermal processes, and wastes disposal reservoirs.

PBDEs are highly persistent in the environment, bioaccumulative in food chain and have a high potential for long-range environmental transport, meaning they can deposit far from their source. These chemicals have been detected in humans and in increasing concentrations in various environmental matrixes, including air, water, soil, sediment, animals and foods in all regions of the world. There is evidence of harmful effects in humans and wildlife, which includes endocrine disruption, immunotoxicity, reproductive toxicity, effects on fetal/child development [10,11,12], thyroid and neurologic function [13], and cancer [14]. Due to the environmental and health concerns, penta-BDE and octa-BDE have been banned in the European Union and voluntarily phased out in the USA since 2004 [15,16]. Recently, the European Commission restricted use of deca-BDE [17]. PBDEs are listed as persistent organic pollutants (POPs) by the Stockholm Convention [18], while deca-BDE (BDE-209) has been classified as a possible human carcinogen by the United States Environmental Protection Agency [8]. Due to the ban of PBDEs, a number of alternative flame retardants has been introduced, which include: organophosphates esters (OPEs) and a range of other brominated and chlorinated novel flame retardants such as tetrabromobisphenol-A (TBBPA), hexabromocyclododecane (HBCD), bis(2,4,6,-tribromphenoxy) ethane (BTBPE), and several phosphate based compounds, such as triphenyl phosphate [19,20,21,22].

Importation of electronics is one major way PBDEs are exported to developing countries such as China [5], India [23], Ghana [24], and Nigeria [25]. When e-wastes are informally recycled using crude methods such as manual dismantling, smelting, and open burning, this leads to incomplete combustion, consequently releasing mixture of hazardous chemicals, including PBDEs, which in turn cause environmental pollution and health problems. PBDE as additive flame retardants are more easily released into the environment than the reactive flame retardants. When released they are attached to particles and transported via various environmental media to distances far from the emission sites. PBDEs enter the environment through multiple pathways, such as emission during manufacturing, from products in use, from combustion, by leaching from landfills, or from recycling of products at the end of their life such as electronics at end-of-life (e-waste) [14]. In the environment, soil and dust are the main environmental receptors of chemical emissions from informal e-waste recycling. Therefore, soils can be secondary sources of emission of PBDEs and soils can contribute to the contamination of air, food and (drinking) water. Hence, soils and dust are the most important environmental media that can reveal the contaminants present in the environment [26]. Moreover, dust is a good indicator for contaminant levels in the atmosphere [27,28,29,30,31].

Data on PBDE concentrations in the environment as a result of informal e-waste recycling in Nigeria is scarce; therefore environmental control on PBDE could be a challenge without adequate information. To gain insight into the PBDE concentrations in the environment as a result of informal e-waste recycling activities we systematically collected soil and dust samples from different selected e-waste recycling sites and from corresponding control sites in three cities where e-waste is recycled in Nigeria. Hence, the objectives of this study were to (1) quantitatively assess the levels of PBDEs in top soils and dust because of different e-waste recycling activities (burning, dismantling, and repair sites); (2) determine the extent to which PBDE concentrations at e-waste sites exceeded the concentrations at control sites; (3) determine the activities that contribute most to the PBDE pollution to the environment; and (4) determine the distribution patterns of the various PBDE congeners. Most importantly, we hope that our findings could be a wake-up call to relevant stakeholders to devise effective interventions to reduce PBDE pollution caused by informal e-waste recycling without impeding the livelihood of the e-waste workers. Our findings are likely to be applicable to other locations or countries where informal e-waste recycling is practiced.

## 2. Materials and Methods

### 2.1. Study Locations and Sites

The methods employed in this study have been well detailed in our previous studies [32]. In brief, this study was conducted in three study locations/cities in Nigeria as depicted in the map as A,B,C, which are Ibadan, Aba, and Lagos respectively. The three study locations are some of the largest cities in Nigeria where e-waste is recycled [25]. In each study location, two e-waste recycling areas were selected. In Lagos, the selected sites were Computer village, Ikeja (6.593° N, 3.342° E) and Alaba international market Ojor (6.462° N, 3.191° E). In Ibadan, the selected sites were Ogunpa (7.383° N, 3.887° E) and Queens Cinema areas (7.392° N, 3.883° E). In Aba, the shopping centre (5.105° N, 7.369° E) and Port-Harcourt Road/Cementary (5.104° N, 7.362° E) and Jubilee road/St Michael’s Road (5.122° N, 7.379° E) were selected ( Figure 1, showing map of the study locations). Alaba international market is the largest market for new and second-hand electronics in West Africa, with approximately ten to fifteen containers arriving daily from Europe and Asia, with each container containing about 400,000 second-hand units [33]. Computer village Ikeja is a popular place where electronics and their parts (new and second hand) can be purchased and repaired. In Ibadan, the selected sites were Ogunpa and Queens Cinema areas. Ogunpa area is known for its activities in scrap/second-hand businesses, including electronics, while Queens cinema is known for sales and repair of both new and second-hand electronics. In Aba, the shopping centre and Port-Harcourt Road/Cementary and Jubilee road/St Michael’s Road were selected. The shopping center area is the biggest market for new and used electronics, while the Port-Harcourt road/Cemetery area is known as an area for scrap/second-hand metal businesses in Aba. In Alaba, Lagos, we found only one big e-waste burning site, which is the largest, oldest, and most studied e-waste burning and dismantling site in Nigeria. In Ogunpa, Ibadan and Cemetery area Aba, the burning sites/spots were much smaller but more spread out in small clusters around the areas.

### 2.2. Study Design

A comparative cross-sectional study design was adopted to gain insights into the PBDE pollution levels at the e-waste recycling sites compared with non-e-waste sites (control sites) in Nigeria. The study is designed in a way that is a representative of other informal e-waste recycling sites. In each study location, a multi-stage random systematic sampling technique was used to include various groups of e-waste workers and e-waste recycling activities (burning, dismantling, and repair) in the selected areas. Soil and dust samples were collected from the selected sites. The control sites were about 500 m away from the e-waste recycling sites, and consisted of areas with reduced human activity such as play grounds, parks, fields, and a university garden. Three types of e-waste recycling activities sites (burning sites, dismantling sites, and repair sites/shops) were analysed.

Top soils (0–10 cm) and various dust samples were collected from the selected e-waste recycling sites. Type of sample collected on each site depends on how feasible it is to collect the sample; this led to an unbalanced design in soil sample collection. At burning sites, only top soil samples were collected, no dust samples were collected at the burning sites because burning activities take place on bare ground. Direct dust from the electronics, mainly from televisions, computers, printers, and air conditioners were also collected. The locations of the sampling spots were georeferenced using a global positioning system application on a phone. Limitations to the methods of sample collection to ensure representative samples were the absence of unpaved surfaces in some place; therefore samples were collected based on feasibility of sample type collection. This study took place in urban areas where most places are paved, these conditions led to an unbalanced number of each sample type. However, this study design is a representative of the sites and major activities at informal e-waste recycling sites. Figure 2 presents a schematic flow diagram of the sample collection from the various e-waste sites in the three study locations. In the previous study by Ohajinwa et al. [32] which was on metal pollution at the same sites, a total of 62 samples. While for this study, a total of 56 samples consisting of 16 top soils (0–10 cm), 29 floor dust, five roadside dust, and six direct dust samples (collected from the inside and outside of electronic devices) were analysed. For the locations, 15, 26, and 29 samples were collected from Aba, Ibadan, and Lagos, see Figure 2. The difference in the number of samples was due to loss of samples and non-detection limits set for the PBDE analysis.

### 2.3. Sample Collection and Preparation

First, the 10ml amber bottles and aluminium foils were treated in the laboratory. The amber bottles were washed with tap water and laboratory detergent, rinsed with a copious amount of tap water, rinsed with distilled water three times, treated with acetone and with hexane, and then oven-dried at 120 °C for 4 h to ensure no traces of POPs were present. Aluminium foils (for sample wraps on the field) were treated with acetone and hexane, then oven dried at 120 °C to ensure no traces of POPs in the aluminum foil.

On the field, for the soil sampling, each selected site was divided into grids of about 2 m to 10 m wide, depending on the size of the site. Samples were systematically collected from three to six points within each site. The samples were bulked together for the top soil to form a composite representative sample for the specific site. Soil samples were collected using a soil auger, and a soil trowel was used to the transfer soil from the soil auger into aluminum foil (sample wraps). To avoid cross contamination, the soil probe/auger and trowel were decontaminated (cleaned first with a brush and wiped thoroughly with wipes) before each sample collection at each sampling site. Dust samples were collected using fiber dusting brushes to gently sweep the dust and collect it with a dustpan. The soil and dust samples were wrapped in a treated aluminum foil, labelled, and transported to the laboratory. A total of 71 samples (56 samples from the e-waste recycling sites and 15 samples from control sites) were analysed. The total set consisted of 22 top soil (0–10 cm depth) samples, 30 floor dust samples, 13 roadside dust samples, and six direct dust samples. Soil and dust samples were air dried for 7 days, avoiding exposure to sunlight. The samples were homogenized, ground with a mortar and pestle, and sieved through a 1 mm mesh sieve to remove bigger particles. Next, they were transferred into individual 10 mL amber bottles, labelled and stored at −20 °C until shipping to the laboratory for analysis. The samples were collected between May and November 2015.

### 2.4. Chemicals and Materials

All the solvents used for extraction, purification and analysis were of high-performance liquid chromatography (HPLC) grade (Spectrum Chemical MFG. Corp., New Brunswick, NJ, USA). Silica gel (100–200 mesh) and neutral aluminum oxide (100–200 mesh) were for chromatography purpose (Sinopharm Chemical Reagent Co., Ltd, Shanghai, China), and they were activated before use (i.e., first washed with hexane/dichloromethane (v/v, 1/1) and then baked at 180 °C for 2 h). Acid silica gel (30% w/w) was prepared with activated silica gel and sulphuric acid before use. Anhydrous sodium sulfate (99% purity) and diatomaceous earth (DE, 100% purity) were purchased from Aladdin Ind. Corp. (Shanghai, China) and ThermoFisher Scientific (Waltham, MA, USA) respectively. They were baked at 400 °C for 4 h before use to remove any traces of organic matter.

A standard mixture solution of 14 PBDE congeners (BDE-COC) PBDEs (BDE-17, BDE-28, BDE-71, BDE-47, BDE-66, BDE-100, BDE-99, BDE-85, BDE-154, BDE-153, BDE-138, BDE-183, BDE-190, and BDE-209) and Individual standards of 4 PBDEs (BDE-77, BDE-206, BDE-207, BDE-208) and PCB-209 were purchased from Accu Standard (New Haven, CT, USA), while Isotopically labeled 13C-PCB-208 was purchased from Cambridge Isotope Laboratories (Tewksbury, MA, USA). We used ^13^C-PCB-208 as the surrogate because, we first used chemical ionization source (CI source) to detect PBDEs with the characteristic ionic fragments, and CI source cannot identify the difference between ^13^C-labeled PBDEs and unlabeled PBDEs. Secondly, ^13^C-PCB-208 can be identified by CI source, and its characteristic ionic fragments contain ^13^C labeled carbon. Also, the physiochemical properties of PCBs and PBDEs are similar with PBDEs.

### 2.5. Sample Extraction and Cleanup

For the PBDE analysis, from each of the samples, 5 g of homogenized sample was thoroughly mixed with 0.6g DE with a mortar and pestle. Each sample was thereafter spiked with 2ng ^13^C- labeled PCB-208 and 10 ng PCB-209 standards, and allowed a static equilibration of 5 min in two cycles. The sample was then extracted using Thermo Scientific Dionex ASE 350 accelerated solvent extraction system with *n*-hexane/dichloromethane (v/v, 1/1) at 90 °C, 1500 psi. After extraction, acid washed copper sheets were added to the extracts to remove sulfur present in the samples. The extracts were evaporated to about 10 mL under a gentle stream of N_2_, and transferred to a conical centrifuge tube. One mL of concentrated sulfuric acid (98%) was added to the concentrated extracts to carbonize part of the impurities present. The supernatants were transferred to a preconditioned glass cleanup column, which was packed with 0.5 cm neutral aluminum oxide, 3.0 cm neutral silica gel, 3.0 cm acid silica gel, and 1.0 cm anhydrous sodium sulfate from the bottom to the top. The columns were then eluted with 20 mL hexane, and the eluent was evaporated to about 0.3 mL and transferred to a 1.5 mL sample vial. After the internal standard (10 ng BDE-77) had been added to the vial, the volume of the solution was made up to 0.5 mL. Many similar previous studies used BDE-77 as internal standard or surrogate [34,35].

### 2.6. Sample Analysis

A 6890 GC/5975 MSD system (Agilent, Santa Clara, CA, USA) operated in negative chemical ion source/ selective ion monitoring (NCI/SIM) mode and equipped with a 15 m DB-XLB column (0.25 mm, 0.1 μm film thickness, J&W Sciebtific, Folsom, CA, USA) was used for PBDE separation and quantification. The samples (1 μL) were injected in split less mode. Helium was used as carrier gas at a flow rate of 1.2 mL/min, and the temperature program was set as follows: 90 °C for 2min, increased to 320 °C at 15 °C/min and held for 7 min. The temperature of GC inlet, transfer line, ionization source and quadrupole were set at 290 °C, 300 °C, and 150 °C. The compounds were monitored at *m*/*z* 79 and 81 for 3–7 brominated BDEs, *m*/*z* 79, 81, 487 and 489 for BDE-206, 207, 208 and 209, *m*/*z* 474, 476 for ^13^C-PCB-208, and *m*/*z* 497.6, 499.6 for PCB-209.

Soil pH was measured using a calibrated pH meter (691, Metrohm AG, Herisau, Switzerland) in a weight: volume ratio of 1:10 of soil and tap water, adopting the USEPA method 9054D [36]. Total organic content (TOC) of the soil and dust was determined as the weight loss of dried soil (3 h at 100 °C) at 550 °C for 5 h [37]. Since PBDEs have a great potential to bind to environmental matrices rich in organic carbon [38], measuring the TOC concentration in the soil and dust samples was used to establish whether there was any correlation between the measured PBDEs and TOC.

### 2.7. Quality Assurance/Quality Control

A meadow soil collected from a cropland in Liaoning, China (123.90° E, 41.38° N) which was tested and demonstrated to be free of most of the studied PBDEs, was used as matrix blank and matrix spike samples. Twenty ng BDE 206, 207, 208, 209 and 4 ng of the other target PBDEs were spiked into 5 g meadow soil to evaluate the method performance. The recoveries for BDE-100, 154, 153, 183, 190, 208, 207, 206 and 209 were 60–107%. For BDE-28, 47, and 99, the recoveries were 32–58%. For all the target compounds, the relative standard deviations of duplicate samples were less than 14%. The recoveries of BDE-17, 66, 71 and 85 were lower than 20%, therefore they were not excluded from the statistically analysis. For the spiked surrogate ^13^C-PCB-208 and PCB-209, the average recoveries in all samples were 71% and 84%, respectively. The method detection limits (MDLs) values characterized as three times the signal-to-noise ratio were 8–164 pg/g for the target PBDEs (Appendix A). The procedural blanks and solvent blanks were analysed simultaneously with samples to check for interferences and contamination. The reported results of PBDEs in the samples were corrected by recoveries of ^13^C-PCB-208. Three criteria were also used to ensure the correct identification of the target compounds [39]: (a) The GC (gas chromatography) retention times matched those of the authentic standards within ±0.1 min. (b) the signal-to-noise ratio was greater than 3:1; and (c) the isotopic ratios between the quantitative and confirmation ions were within ±15% of the theoretical values. Both ^13^C-PCB-208 and PCB-209 were used as surrogate to indicate the stability of the recoveries of each sample. In addition, this measure helps to monitor the recoveries of the target compounds at different concentration level.

### 2.8. Data Analysis

A two-tailed Pearson correlation coefficient was used to determine the strength of association between PBDE congeners and Total Organic Carbon (TOC). In this contribution, we considered 13 PBDE congeners –BDE-28, BDE-47, BDE-100, BDE-99, BDE-153, BDE-154, BDE-138, BDE-183, BDE-190, BDE-207, BDE-208, BDE-206, BDE-209. The PBDE concentrations were summarized using descriptive statistics (frequencies, median means, and standard deviations). In addition, the pollution status of the different sites were evaluated by calculating the ratio of various PBDE concentrations at the e-waste sites compared to the control sites, which is known as exceedance (Ex) or contamination factor (CF). A CF < 1 indicates low contamination, 1 < CF < 3 moderate contamination, 3 < CF < 6 considerable contamination and CF > 6 indicates a very high contamination level [40,41].

To understand variabilities in the mean concentration distribution of PBDEs in the environmental samples, we evaluated the differences in the mean concentrations of the individual PBDE congeners and the sum of the PBDE congeners (∑_13_PBDE) by running a series of one-way (using activity site as factor on the log transformed data for soil, floor dust, roadside dust and direct dust separately. Additional series of two-way ANOVAs were run, firstly with activity sites (burning, dismantling, repair, and control sites) and location as explanatory variables; secondly with type of activity/activity site and type of sample (soil and dust) as explanatory variables. Bonferroni post-hoc tests were included to interpret the significant main effects of the ANOVA outputs. Principal component analysis (PCA) was used to evaluate whether variation in PBDE concentrations was similar across PBDE compounds. A *p*-value of 0.05 was considered statistically significant. All statistical analyses were performed using SPSS version 23 (IBM Statistics20, IBM, Armonk, NY, USA).

## 3. Results

### 3.1. Physicochemical Characteristics of the Soil and Dust Samples

The soil texture in Lagos and Aba is sandy loam and in Ibadan is sandy clay [42]. The physiochemical characteristics of the soil and dust samples such as pH, total organic matter content (TOC), are presented on Appendix A. The pH at the burning sites ranged from 7.2 to 8.4, dismantling sites (1.9 to 9), repair sites (7.9 to 9.24), and control sites (7.4 to 9) while the TOC for the burning sites ranged from 8 to 36.3%, dismantling sites (1.6 to 24%), repair sites (1.14 to 24%), and control sites (0.98 to 5.3%). There were significant differences in the pH (*p* = 0.004) and TOC (*p* = 0.000) levels between all the sites. Also, there were significant differences in the pH (*p* = 0.006) and TOC (*p* = 0.00) between the various e-waste recycling sites.

### 3.2. PBDE Concentrations at the Various Activity Sites

PBDE concentrations at e-waste sites and control sites are presented in Appendix A and in Figure 3. PBDE congeners were detected in all sampling sites, indicating that PBDEs were widespread pollutants in this research area. The total concentration (∑_13_PBDE) ranged from 1.702 to 149,770.560 ng/g. The most abundant PBDE congener in all the sites and samples was BDE-209, with concentrations ranging from 0.850 to 147,091.400 ng/g. The maximum ∑_13_PBDE was found in direct dust from TV repair shops in Ibadan. This high variability in PBDE concentrations is a reflection of the activities on the sites. The abundance of the PBDE congeners (considering the median of all the samples) in all the locations are generally in this order: BDE-209 > BDE-207 > BDE-206 > BDE-183 > BDE-208, BDE-99 > BDE-153 > BDE-47 > BDE-190 > BDE-154 > BDE-100 >BDE-28 > BDE-138 >. The PBDE concentrations at the e-waste exceeded the control sites by many folds, see Figure 3 and Appendix A for details of the exceedance (EX) levels. The general pattern of the PBDEs distribution at the e-waste sites showed concentrations in this decreasing order: burning sites > dismantling sites > repair sites > control sites. This shows that burning activities contribute most to the PBDE concentrations in the environmental matrices, as is also clear from the patterns in ∑_13_PBDE (Figure 3). In all the sample types, the highest concentrations were found in Lagos.

### 3.3. Assessment of the Top Soil Samples

The one-way ANOVAs showed a significant difference in the concentration of ∑_13_PBDE and all the PBDE congeners, except BDE-28, in top soils between the activity sites, (*p* = 0.05). Post-hoc comparisons indicated the main differences is between control and burning sites, and control and dismantling sites for all the PBDE congeners and ∑_13_PBDE, see Figure 4 and Appendix A.

While the type of activities on the sites influences the PBDE concentrations on the sites, that effect might differ across locations. To test this assumption, a two-way ANOVA was performed and this ANOVA confirmed a significant difference in all the PBDE concentrations between the activity sites, with the burning sites having the highest concentrations, followed by dismantling sites, then control sites. Forty-one to 100% of the variability in PBDE concentrations was accounted for by the activities at the sites. However, there was no significant difference in the concentration of any of the PBDE congeners between locations, except for BDE-28, 208, and 207. The PBDE concentrations were generally highest in Aba, followed by Lagos, then Ibadan. This excludes BDE-28, which is highest in Lagos. Also, the interaction of activities at the sites and location showed no significant difference on any of the PBDE congeners and ∑_13_PBDE concentrations, except for BDE-28 (see Appendix A).

#### 3.3.1. Assessment of the Floor Dust Samples

A one-way ANOVA showed a significant difference (*p* = 0.05) in the concentration of some of the PBDE congeners (BDE-99, 47,100, 154, 190, 138) in floor dust between dismantling and repair sites (Figure 5).

The two-way ANOVAs testing individual PBDE congeners and total PBDE (∑_13_PBDE) concentrations in floor dust from the activity sites (dismantling and repair sites) and across the locations (Lagos, Ibadan, and Aba), showed significant differences in the concentration of some PBDE congeners (except BDE-47, 100, 99, 154, 138, 190) between the activity sites, with dismantling sites having higher concentrations than repair sites. Also, there were significant differences in the concentration of some PBDE congeners (BDE-47, 100, 99) between locations, with Ibadan having generally the highest PBDE concentrations, followed by Lagos, then Aba. The interactions of activities at the sites and location showed no significant difference on any of the PBDE congeners and ∑_13_PBDE concentrations, except for BDE-154 (see Appendix A).

#### 3.3.2. Assessment of the Roadside Dust Samples

Roadside dust samples were collected only from Lagos and Aba. One-way ANOVA showed a significant difference (*p* = 0.05) in the concentration of almost all the PBDE congeners in roadside dust between dismantling and control sites (Figure 6).

#### 3.3.3. Assessment of the Direct Dust Samples

Direct dust samples from electronics were collected from dismantling and repair sites from Ibadan only. A one-way ANOVA showed no significant difference in the concentration of any of the PBDE congeners in direct dust between dismantling and repair sites, except BDE-100, (Figure 7).

### 3.4. Patterns in PBDEs Contamination

The principal component analysis (PCA) using direct oblimin rotation was performed on the correlation matrix of the PBDE concentrations to establish whether the contaminants were actually arising from the same source or not. The analysis revealed one common axis of variation in PBDE concentrations, which accounted for 84% of the total variance. All PBDEs varied in the same direction (Figure 8). All the PBDE congeners had high positive loadings of ≥0.803. These findings indicate that PBDEs contamination has one common driver, which might suggest on common source. Pearson correlation confirmed that all PBDE congeners at the e-waste sites strongly correlated positively with each other. There was also correlation between the TOC and all the individual PBDEs and ∑_13_PBDE (Appendix A).

## 4. Discussion

In this study we analysed the PBDE concentrations in soil and dust samples from different e-waste activity sites (burning, dismantling, and repair sites) as compared to corresponding control sites in three different cities in Nigeria. The strengths of this study are the analyses of the interrelationships between the e-waste sites and the environmental matrices such as top soils, floor dusts, roadside dust, and direct dust from the electronics in which PBDEs are measured. These distinctions, we did not find in the previous studies.

Most e-waste recycling activities, especially at dismantling and burning sites are carried out outdoors. The recycling activities include storage, washing, cleaning, dismantling, and metal recovery through stripping of wires or open burning. The remains of *e*-waste materials from the recycling activities are dumped outside on the ground. Most repair activities, which involve soldering of various parts, take place indoors but also sometimes outdoors, depending on the settings of the work environment and the weather condition. These activities release large quantities of hazardous substances. Soils and dusts are a major repository for pollutants released into the environment by human activities, and they are important environmental media that can provide information about the level, distribution, and fate of contaminants present in the environment as a result of informal *e*-waste recycling.

Despite the increasing volumes of e-waste generated over the years, collection and recycling of e-waste are still not improved in developing countries [27]. Nigeria imports the largest volume of new and used electronic and electrical equipment in Africa [25]. The amount of e-waste generated in Nigeria has increased from 219 kilotonnes in 2014 [1] to 277 kilotonnes in 2016 [2]. This increase is despite the high weight reduction of electronic devices like computers (PCs). Almost all the e-waste generated are recycled in an unsafe/informal manner [1,25]. This situation in Nigeria is likely to be representative for informal e-waste recycling in countries that lack the resources for safe e-waste recycling such as in India, Brazil, Mexico [43], and Ghana [44] among others.

### 4.1. Extent of Pollution As A Result of Informal E-waste Recycling

Our findings revealed that open burning of e-waste is the most polluting e-waste recycling activity. This is in accordance with a study by Matsukami et al., [45] which compared burning sites and other e-waste processing sites in Vietnam. PBDEs do not occur naturally in the environment, but traces of PBDEs were found in control sites, indicating deposits of PBDEs in the environment not too far from e-waste recycling sites. These PBDEs might have been transported by wind/air to nearby vicinities, which is in agreement with the observations of decreasing concentrations of PBDEs with increasing distance from e-waste sites [46,47].

In this study we included dust samples, considering that some of the activities (such as repair activities) do not take place on soil (bare ground) most of the time. The added value of determining the PBDE concentrations in the dust samples was to ensure that different types of e-waste activity sites were studied. Dust is one of the main sources of exposure to PBDEs via inhalation or ingestion. Thus it provides information about the level of contaminants in the indoor atmosphere as well as the levels of contaminants to which the workers and the public are exposed. Dust also reflects the characteristics of short and long term activities in the area. Furthermore, the combination of soil and dust samples gives a comprehensive overview of the impact of informal e-waste recycling on the environment. It is likely that there is a cross transference of the PBDEs from the floor dust (indoor) from the shops to the soil, and from the soil from the burning and dismantling sites into the shops (contaminating floor dust). There is also a probable transfer of PBDEs from the e-waste sites to locations farther away from the e-waste recycling sites, which is in agreement with the previous findings [48,49].

We found high concentrations of PBDE congeners at the e-waste sites with the higher molecular weight PBDEs (BDE-209, BDE-153, BDE-183) having the highest concentrations, and BDE-209 being the most abundant. Predominance of BDE-209 in samples is probably due to the fact that the deca-BDE mixture is the predominant PBDE still in use [20,50]. This is in agreement with the results of previous studies in Turkey [51], in five Asian countries [52], in Vietnam by Matsukami et al., (2017) [45], in Ghana by Akortia et al., (2017) [44], and in Nigeria and China [53]. This is similar to the findings of Takigami et al. [54] showing the highest concentrations of PBDEs (BDE-209) in dust from e-waste sites. The maximum concentration of BDE-209 was 147,091 ng/g, as found in dust from a television. However, in a similar study in Ghana, BDE-28 was found to have the highest concentration instead of BDE-209 [44].

The PBDE levels present at the e-waste sites and the control sites reflects the pollution from anthropogenic sources in urban areas. Comparing the mean concentrations of BDE-209 levels in top soils to those reported in previous studies at the same study areas, it is found that BDE-209 concentrations are decreasing slowly in a space of three years by a factor of 1 and 3 at Alaba and Computer village respectively (Table 1). 

It is important to note that the e-waste dismantling and burning sites at Alaba in 2012 is different from the site in 2015 because the site was relocated within Alaba. We suspect that if the same site was retained in Alaba since inception of *e*-waste burning at Alaba, the levels of the PBDE congeners maybe higher than the levels detected. At the same time, we gave it a thought that the decrease may also be because PBDEs use in electronics has been banned. The median BDE-209 concentration at the burning sites is as high as 17,587 ng/g at Alaba, Lagos (Appendix A). When compared to Guiyu, China, the BDE-209 levels at Guiyu decreased by a factor of 10 in a space of two years (Table 1).

PBDE levels found at Alaba sites are higher than the levels found in the widely studied area of Guiyu, China in 2014 and 2015. Guiyu, China is known for its notorious intensive unregulated crude e-waste recycling activities. This shows that maybe more notorious e-waste recycling activity maybe going on somewhere in Nigeria, which are yet to be reported. These findings further show that the PBDE concentrations in urban cities (mostly in the slums) in Nigeria are still high and call for concern. This consequently implies that more people in the general population (besides e-waste workers) might be exposed to PBDE. This is more disturbing as majority of the e-waste workers are unaware of the health risks associated with their jobs [48] and do not use any form of PPE [49] or take appropriate cause to protect their health or the environment.

In this study we distinguished between PBDE levels at various e-waste recycling sites (burning, dismantling, and repair sites). These distinctions were not made in the other studies. The different e-waste activities had significantly different mean concentrations of PBDEs. The post-hoc tests revealed that the biggest effect is seen at the burning and control sites being significantly different from the other sites. There was no significant difference in the PBDE concentrations of similar activity sites between the locations, and between sample types for most of the PBDEs. These findings indicate that activities in the vicinity have impact on the level of PBDEs in an area. All PBDE congeners positively correlated with each other. The positive correlations between the PBDE congeners indicate that the PBDEs are likely from the same source with similar emission patterns. This was further confirmed by the PCA, in which 84% of the total variance of all PBDE congeners was accounted for by a common axis.

Although penta-BDE, octa-BDE, and deca-BBE are banned in developed countries where electronics are manufactured, and despite Nigeria having regulations on e-waste management which in turn controls PBDE emissions, PBDEs were found in high concentrations at the e-waste sites and at the control sites, this shows that PBDEs are ubiquitos in Nigerian environment, as stated in other studies which detected PBDE in various environmental matrices [55,56,57,58]. There is a possiblity that higher molecular weight BDEs debrominates to lower molecular weight BDEs as stated by Zhang et al. [59]. We found deca-BDE (BDE-209), a high molecular weight BDEs having the highest concentrations at all sites. Therefore deca-BDE may represent important reservoir lower-PBDE congeners. Generally, there were positive correlations between all the PBDE congeners. There were also positive correlations between all the PBDEs and the TOC in top soils from e-waste (Appendix A). Correlation of TOC with PBDE suggests that PBDE binds to environmental matrix rich in organic carbon. As PBDEs bind strongly to soil particles, they may remain in soil for several years or even decades. Total Organic matter content (TOC) influences the distribution of PBDE in the soil and dust to some extent.

### 4.2. Implications of High PBDE Concentrations on Health and Environment

As PBDEs do not naturally occur in the environment, there is no doubt that e-waste recycling is a major source of PBDE pollution in Nigeria. When released, they bind strongly (especially congeners with higher content of bromine bind more strongly) to soil, sediment particles, and sewage sludge, in turn making them less mobile in the environment. Therefore, they bioaccumulate and biomagnify in aquatic organisms, fish, and plants, and are eventually transferred up the food chain, ultimately to humans [57]. Moderate to high PBDE congeners are found in air samples closer to the source of pollution, while PBDE congeners with less bromine atoms travel greater distances from their same source [58,59], meaning that people living far away from the source of release may also be at risk of exposure to PBDE. It is assumed that the higher PBDEs may degrade to lower PBDE congeners like tetra-, penta-, and hexa-BDEs in the environment, and that the PBDEs with lesser bromine atoms are more persistent in the atmosphere [60]. 

The ∑_13_PBDE concentrations found in soils and various dust samples exceeded the Agency for Toxic Substances and Disease Registry (ATSDR) oral Minimal Risk Levels (MRL) of 0.00006 mg/kg/day for lower-brominated PBDEs based on a LOAEL (lowest-observed-adverse-effect level) for endocrine effects in rats. Our values also exceeded The EPA’s reference doses (RfDs) for penta, octa-, and decaBDEs are 2 × 10 ^−3^, 3 × 10 ^−3^, and 7 ×10 ^−3^ mg/kg/day, respectively [61], suggesting that PBDEs could adversely affect animals and other sensitive species in the environment, and consequently humans in and around the study areas. Humans can be exposed to PBDEs and metals though inhalation, dermal absorption, and consumption of contaminated foods such as fish, meat, and dairy products [62,63]. This is a considerable environmental concern and most likely a health concern. We recommend further toxicological studies on the e-waste workers. It is hoped that the results of this study are a wake-up call on the need for more effective strategies on enforcement of e-waste regulations in Nigeria. We recommend that the enforcement would be effective if the regulations are made through the lens of the informal sector and enforcement agencies collaborates with the informal sector so as not to impade the workers’ livelihood. The findings in this study is representative of what might be going on in other places unsafe e-waste recycling is practice, therefore solutions proferred for Nigeria is applicable to other places.

## 5. Conclusions

Our study showed that PBDE concentrations at the e-waste recycling sites were elevated compared to those detected at the control sites by 100 s to 1000 s times, with BDE-209 being the most abundant in all the samples and at all the sites. There was a significant difference in concentrations of PBDEs at the various e-waste activities sites in this decreasing order: burning sites > dismantling sites > repair sites > control sites. This proves that the type of activities at the sites influences the level of PBDEs, with burning activities having the most effect. This study demonstrates that crude recycling of e-waste contributes significantly to emissions of organic pollutants in the environment. Comparing our results with past studies in the same locations, not much has changed in the the PBDE concentrations considering that PBDE use in electronics has been banned, suggesting that the situation calls for urgent action. Our results suggest that the informal e-waste recycling has negative impacts on the enviroment and consequently on health. There is an urgent need for more effective actions to stop open burning of e-waste and to reverse or stop the environmental deterioratation as a result of informal e-waste recycling. One way to stop unsafe recycling of e-waste is to adopt a bottom-up approach in stopping these unsafe practices by: (1) for the formal institutions to appreciate and work with the informal sectors (2) to the create awearness on the potential health risks of unsafe recycling of e-waste among th e-waste workers. 

## Figures and Tables

**Figure 1 ijerph-16-00360-f001:**
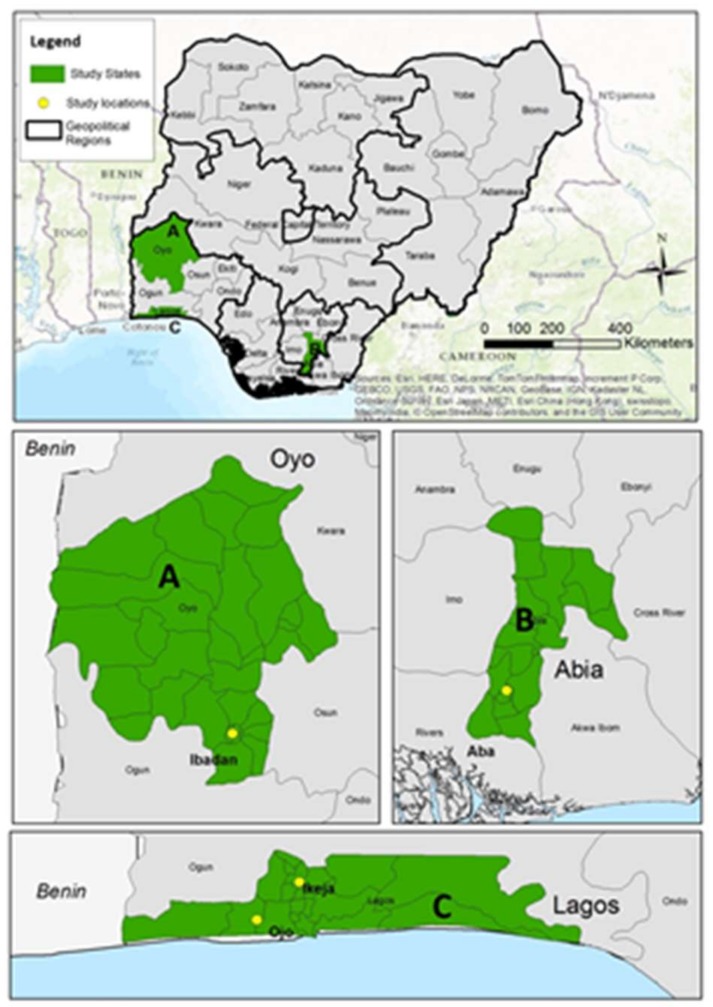
Map of Nigeria showing the study locations.

**Figure 2 ijerph-16-00360-f002:**
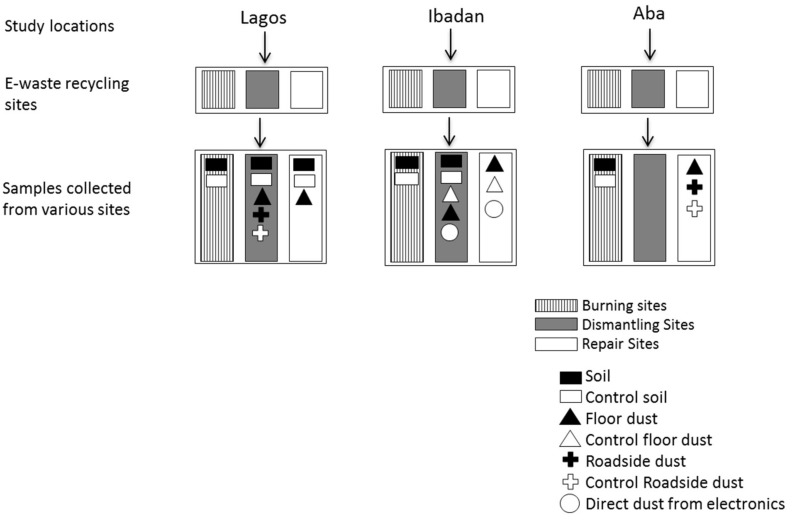
Schematic flow diagram of sample collection in the study locations.

**Figure 3 ijerph-16-00360-f003:**
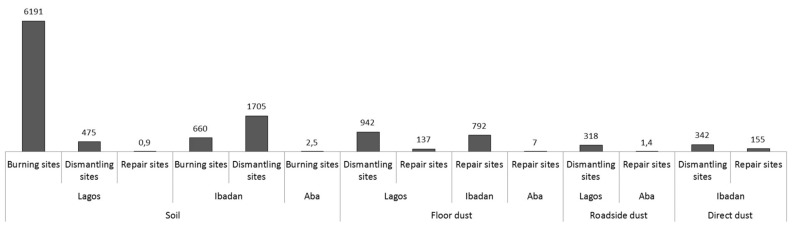
Exceedances of ∑_13_PBDE concentrations in the samples from the e-waste recycling sitescompared to the control sites across locations.

**Figure 4 ijerph-16-00360-f004:**
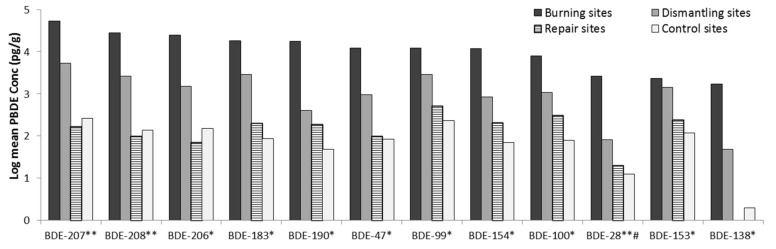
PBDE concentrations in top soils across the sites. The PBDE congeners influenced by activity at the sites are indicated with *, and an additional * for those influenced by location, and # for interaction between activity and location.

**Figure 5 ijerph-16-00360-f005:**
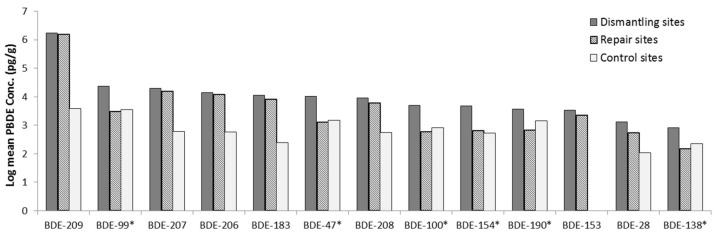
PBDEs concentration in floor dust across the sites. The PBDE congeners influenced by activity at the sites are indicated with *.

**Figure 6 ijerph-16-00360-f006:**
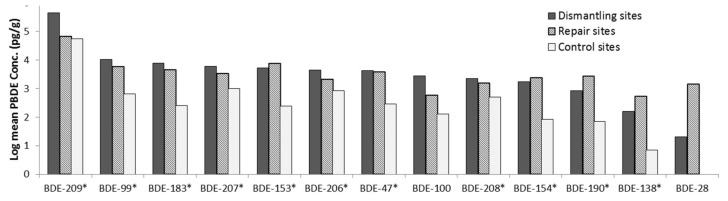
PBDE concentration in roadside dust across the sites. The PBDE congeners influenced by activity at the sites are indicated with *.

**Figure 7 ijerph-16-00360-f007:**
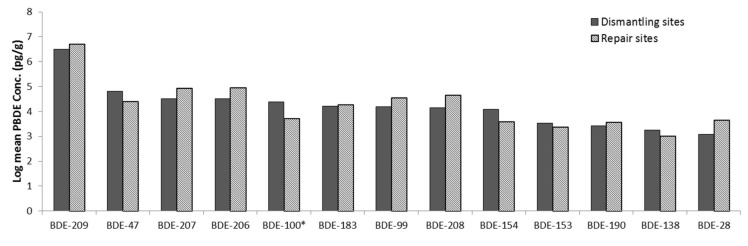
PBDE concentration in direct dust from electronics at the sites. Generally, there were no significant differences in the PBDE concentrations, except for BDE-100.

**Figure 8 ijerph-16-00360-f008:**
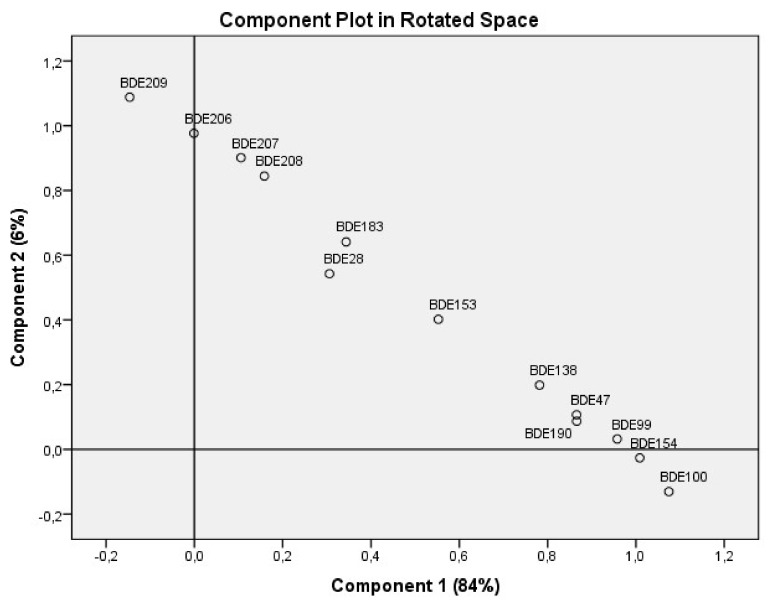
PCA plot of PBDE concentrations.

**Table 1 ijerph-16-00360-t001:** Comparison of PBDE concentrations in soils at e-waste sites with other studies.

Countries	Units	BDE-209	Reference
Computer village Nigeria	ng/g dry wt	583	This study
Alaba international market, Nigeria	ng/g dry wt	7648± 8369	
Soil at e-waste recycling site, Ghana	ng/g dry wt	10.6 ± 16.6	[44]
Soils near e-waste recycling site, China	ng/g dry wt	3400 ± 4200	[50]
Soils near e-waste recycling site South Korea	ng/g dry wt	8.8 ± 11	
Soils near e-waste recycling site Vietnam	ng/g dry wt	63	
Guiyu Soils near e-waste recycling site, China	ng/g dry wt	1157 ± 1131	[55]
Computer village dumpsite soil, Nigeria	ng/g dry wt	1820	[51]
Alaba international market dumpsite soil, Nigeria	ng/g dry wt	9800	
Guiyu e-waste dumpsite soil, China	ng/g dry wt	12,130	

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
