# Peer review of "Hydrophobic Organic Pollutants in Soils and Dusts at Electronic Waste Recycling Sites: Occurrence and Possible Impacts of Polybrominated Diphenyl Ethers"

_ijerph, 2019, doi:10.3390/ijerph16030360_

Round 1
Reviewer 1 Report
The paper covers an important topic and the policy implications can be very useful for other countries. The author measured the concentrations of 13 PBDE congeners, in top soils (0-250px) and in various dust samples from different e-waste recycling sites (burning, dismantling, repair). The paper has potential, however some changes must be provided before the paper can be considered for publication, namely:
· The literature review can be improved;
· The Introduction could also be improved, for example, in Europe, the EPR is not only relevant for WEEE but for other flows (vide Cruz et al. 2015);
· It could be included a paragraph presenting the organization of the paper;
· Moreover, more information can be included about the case study (e.g. the GDP, …);
· All the abbreviations must be presented in the text;
· Regulation is a relevant matter in waste sector and it should be developed (see Simões and Marques, 2012);
· The methodology adopted can be better justified (e.g. including the main advantages and limitations);
· A table for statistical analysis of the data used would be relevant to be included in the chapter;
· All formulas and expression must be numbered and the variables must be presented in the text;
· All sources of information must be presented;
· Units of the variable have to be indicated;
· Try to add statistical analysis to the results;
· I would expect more recommendations for decision makers;
· Conclusions must be improved;
· The references must be homogenized and in line with the Journal Guidelines (for example, some papers do not indicate the issue).
References:
CRUZ, N. et al. (2014). Packaging waste recycling in Europe: is the industry paying for it?. Waste Management. Vol. 34, no. 2, pp. 298-308.
SIMÕES, P.; MARQUES, R. (2012). Influence of regulation on the productivity of waste utilities. What can we learn with the Portuguese experience? Waste Management. Vol. 32, no. 6, pp. 1266-1275.
Author Response
The paper covers an important topic and the policy implications can be very useful for other countries. The author measured the concentrations of 13 PBDE congeners, in top soils (0-250px) and in various dust samples from different e-waste recycling sites (burning, dismantling, repair). The paper has potential, however some changes must be provided before the paper can be considered for publication, namely:
Reply: Thank you for the encouragement.
· The literature review can be improved;
·The Introduction could also be improved, for example, in Europe, the EPR is not only relevant for WEEE but for other flows (vide Cruz et al. 2015);
Reply: We have included the extension of EPR to other types of waste and the suggested paper by Cruz et al. is referenced, see line 50-56
It could be included a paragraph presenting the organization of the paper;
Reply: I could not do this because including table of content did not flow in the paper, besides, the table of contents is clearly outlined
·Moreover, more information can be included about the case study (e.g. the GDP, …);
Reply: This is not clear, The word GDP was never used in the paper
·All the abbreviations must be presented in the text;
Reply: The full meaning of all the abbreviations, the missing (GC) abbreviation has been included, see line 273 (GC= Gas chromatography)
·Regulation is a relevant matter in waste sector and it should be developed (see Simões and Marques, 2012);
Reply: Thank you for bringing this paper to my knowledge, I have read it and made the necessary adjustments. However, this paper is about various local regulations to improve solid waste management. In this paper, we focused on the impact of PBDE pollutions at the various e-waste sites
·The methodology adopted can be better justified (e.g. including the main advantages and limitations);
Reply: Thank you for the observation. We have included the limitations and strengths of the paper, see line 169 - 173 and line 393 - 397
·A table for statistical analysis of the data used would be relevant to be included in the chapter;
Reply: The table of statistics is already appended as a supplementary information, The table of statistics is too long, it covers 13 PBDE congeners from 3 different locations, 3 different e-waste recycling sites, 4 different sample types. They are all adequately referred to in the text.
· All formulas and expression must be numbered and the variables must be presented in the text;
Reply: This is not clear, formulas were not used in this paper.
·All sources of information must be presented;
Reply: All the statements have been adequately referenced.
·Units of the variable have to be indicated;
Reply: The units to the concentrations of PBDE are inserted on the title of each table and also properly mentioned in the text.
·Try to add statistical analysis to the results;
Reply: We used one-way and two-way ANOVAs to analysed the results and it is stated in the text, and data on the supplementary.
· I would expect more recommendations for decision makers;
Reply: Thank you for observation. We have included more recommendations, see line 534 - 539.
·Conclusions must be improved.
Reply: Including more recommendation has improved the conclusions. Also, information on the order of intensity of PBDE pollution has been included, see line 526 - 528.
·The references must be homogenized and in line with the Journal Guidelines (for example, some papers do not indicate the issue).
Reply: Thank you for this observation. We have corrected all the references in line with the journal's guidelines.
Reviewer 2 Report
The paper has compared the PBDEs concentration of e-waste sites in Nigeria. PBDEs is one of persist organic pollutant and very toxic. This study may contribute to further stay of PBDEs. However, the paper has some issue.
1. The methodology part is too long some of section could be merged. For example, 2.1 and 2.2. Sample analysis part could use a table to reduce the words.
2. The author could discuss the relation between different sample. For example, soil, floor dust and roadside dust samples were collected in Lagos. The author could further discuss the PBDEs concentration with distance.
3. The reference style need to meet with the journal standard.
Author Response
The paper has compared the PBDEs concentration of e-waste sites in Nigeria. PBDEs is one of persist organic pollutant and very toxic. This study may contribute to further stay of PBDEs. However, the paper has some issue.
Reply: Thank you for the encouragement
1. The methodology part is too long some of section could be merged. For example, 2.1 and 2.2. Sample analysis part could use a table to reduce the words.
Reply: Sub-sections 2.1 (description of study locations and sites) and 2.2 (description of study design) address different parts of the method section. From the previous experiences and publications, it was important that the context of the research is clear. This is to ensure that the sampling was done scientifically to ensure repeatability. Also, it was important to show that the samples are a representative of the locations studies and that results can be applied in similar situations.
2. The author could discuss the relation between different sample. For example, soil, floor dust and roadside dust samples were collected in Lagos. The author could further discuss the PBDEs concentration with distance.
Reply: We did not collect samples based on distances in the same location. Different types of samples were collected based on the feasibility of sample collection at the e-waste recycling sites, see line 163 - 164. For example, at the repair sites/workshops, the floor of the offices are paved or floored, therefore soil samples cannot be collected at the offices. There were no special relationships between the different sample types.
3. The reference style need to meet with the journal standard.
Reply: Thank you for this observation. We have corrected the references to meet the journal's standard.